# Unsupervised Learning of 3D Structure from Images

**Danilo Jimenez Rezende\***
danilor@google.com

**S. M. Ali Eslami\***
aeslami@google.com

**Shakir Mohamed\***
shakir@google.com

**Peter Battaglia\***
peterbattaglia@google.com

**Max Jaderberg\***
jaderberg@google.com

**Nicolas Heess\***
heess@google.com
\* Google DeepMind

## Abstract

A key goal of computer vision is to recover the underlying 3D structure that gives rise to 2D observations of the world. If endowed with 3D understanding, agents can abstract away from the complexity of the rendering process to form stable, disentangled representations of scene elements. In this paper we learn strong deep generative models of 3D structures, and recover these structures from 2D images via probabilistic inference. We demonstrate high-quality samples and report log-likelihoods on several datasets, including ShapeNet [2], and establish the first benchmarks in the literature. We also show how these models and their inference networks can be trained jointly, end-to-end, and directly from 2D images without any use of ground-truth 3D labels. This demonstrates for the first time the feasibility of learning to infer 3D representations of the world in a purely unsupervised manner.

## 1 Introduction

We live in a three-dimensional world, yet our observations of it are typically in the form of two-dimensional projections that we capture with our eyes or with cameras. A key goal of computer vision is that of recovering the underlying 3D structure that gives rise to these 2D observations.

The 2D projection of a scene is a complex function of the attributes and positions of the camera, lights and objects that make up the scene. If endowed with 3D understanding, agents can abstract away from this complexity to form stable, disentangled representations, e.g., recognizing that a chair is a chair whether seen from above or from the side, under different lighting conditions, or under partial occlusion. Moreover, such representations would allow agents to determine downstream properties of these elements more easily and with less training, e.g., enabling intuitive physical reasoning about the stability of the chair, planning a path to approach it, or figuring out how best to pick it up or sit on it. Models of 3D representations also have applications in scene completion, denoising, compression and generative virtual reality.

There have been many attempts at performing this kind of reasoning, dating back to the earliest years of the field. Despite this, progress has been slow for several reasons: First, the task is inherently ill-posed. Objects always appear under self-occlusion, and there are an infinite number of 3D structures that could give rise to a particular 2D observation. The natural way to address this problem is by learning statistical models that recognize which 3D structures are likely and which are not. Second, even when endowed with such a statistical model, inference is intractable. This includes the sub-tasks of mapping image pixels to 3D representations, detecting and establishing correspondences between

different images of the same structures, and that of handling the multi-modality of the representations in this 3D space. Third, it is unclear how 3D structures are best represented, e.g., via dense volumes of voxels, via a collection of vertices, edges and faces that define a polyhedral mesh, or some other kind of representation. Finally, ground-truth 3D data is difficult and expensive to collect and therefore datasets have so far been relatively limited in size and scope.

In this paper we introduce a family of generative models of 3D structures and recover these structures from 2D images via probabilistic inference. Learning models of 3D structures directly from pixels has been a long-standing research problem and a number of approaches with different levels of underlying assumptions and feature engineering have been proposed. Traditional approaches to vision as inverse graphics [20, 17, 19] and analysis-by-synthesis [23, 27, 16, 28] rely on heavily engineered visual features with which inference of object properties such as shape and pose is substantially simplified. More recent work [16, 4, 3, 30] addresses some of these limitations by learning parts of the encoding-decoding pipeline depicted in figure 2 in separate stages. Concurrent to our work [10] also develops a generative model of volumetric data based on adversarial methods. We discuss other related work in A.1. Unlike existing approaches, our approach is one of the first to learn 3D representations in an unsupervised, end-to-end manner, directly from 2D images.

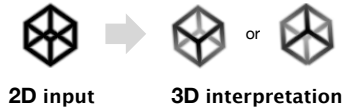

Figure 1: **Motivation:** The 3D representation of a 2D image is ambiguous and multi-modal. We achieve such reasoning by learning a generative model of 3D structures, and recover this structure from 2D images via probabilistic inference.

Our contributions are as follows. (a) We design a strong generative model of 3D structures, defined over the space of volumes and meshes, combining ideas from state-of-the-art generative models of images [7]. (b) We show that our models produce high-quality samples, can effectively capture uncertainty and are amenable to probabilistic inference, allowing for applications in 3D generation and simulation. We report log-likelihoods on a dataset of shape primitives, a 3D version of MNIST, and on ShapeNet [2], which to the best of our knowledge, constitutes the first quantitative benchmark for 3D density modeling. (c) We show how complex inference tasks, e.g., that of inferring plausible 3D structures given a 2D image, can be achieved using conditional training of the models. We demonstrate that such models recover 3D representations in one forward pass of a neural network and they accurately capture the multi-modality of the posterior. (d) We explore both volumetric and mesh-based representations of 3D structure. The latter is achieved by flexible inclusion of off-the-shelf renders such as OpenGL [22]. This allows us to build in further knowledge of the rendering process, e.g., how light bounces of surfaces and interacts with its material's attributes. (e) We show how the aforementioned models and inference networks can be trained end-to-end directly from 2D images without any use of ground-truth 3D labels. This demonstrates for the first time the feasibility of learning to infer 3D representations of the world in a purely unsupervised manner.

## 2 Conditional Generative Models

In this section we develop our framework for learning models of 3D structure from volumetric data or directly from images. We consider conditional latent variable models, structured as in figure 2 (left). Given an observed volume or image $\mathbf{x}$ and a context $\mathbf{c}$, we wish to infer a corresponding 3D representation $\mathbf{h}$ (which can be a volume or a mesh). This is achieved by modelling the latent manifold of object shapes and poses via the low-dimensional codes $\mathbf{z}$. The context is any quantity that is always observed at both train- and test-time, and it conditions all computations of inference and generation (see figure 2, middle). In our experiments, context is either 1) nothing, 2) an object class label, or 3) one or more views of the scene from different cameras.

Our models employ a generative process which consists of first generating a 3D representation $\mathbf{h}$ (figure 2, middle) and then projecting to the domain of the observed data (figure 2, right). For instance, the model will first generate a volume or mesh representation of a scene or object and then render it down using a convolutional network or an OpenGL renderer to form a 2D image.

Generative models with latent variables describe probability densities $p(\mathbf{x})$ over datapoints $\mathbf{x}$ implicitly through a marginalization of the set of latent variables $\mathbf{z}$, $p(\mathbf{x}) = \int p_\theta(\mathbf{x}|\mathbf{z})p(\mathbf{z})d\mathbf{z}$. Flexible models can be built by using multiple layers of latent variables, where each layer specifies a conditional distribution parameterized by a deep neural network. Examples of such models include [12, 15, 24]. The marginal likelihood $p(\mathbf{x})$ is intractable and we must resort to approximations.

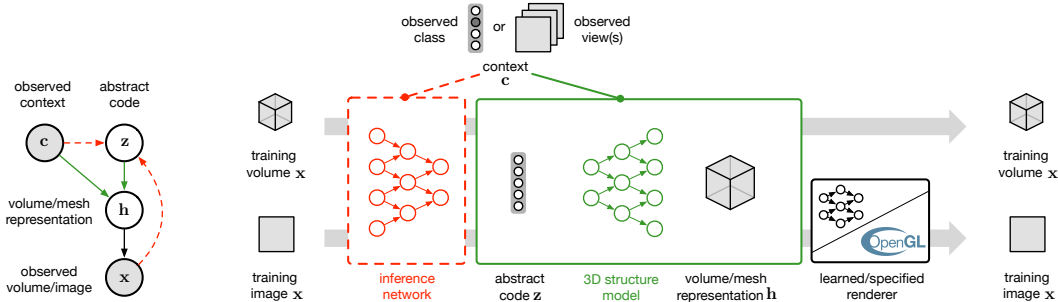

Figure 2: **Proposed framework:** *Left:* Given an observed volume or image $\mathbf{x}$ and contextual information $\mathbf{c}$, we wish to infer a corresponding 3D representation $\mathbf{h}$ (which can be a volume or a mesh). This is achieved by modeling the latent manifold of object shapes via the low-dimensional codes $\mathbf{z}$. In experiments we will consider unconditional models (i.e., no context), as well as models where the context $\mathbf{c}$ is class or one or more 2D views of the scene. *Right:* We train a context-conditional inference network (red) and object model (green). When ground-truth volumes are available, they can be trained directly. When only ground-truth *images* are available, a renderer is required to measure the distance between an inferred 3D representation and the ground-truth image.

We opt for variational approximations [13], in which we bound the marginal likelihood $p(\mathbf{x})$ by $\mathcal{F} = \mathbb{E}_{q(\mathbf{z}|\mathbf{x})}[\log p_\theta(\mathbf{x}|\mathbf{z})] - \mathrm{KL}[q_\phi(\mathbf{z}|\mathbf{x})\|p(\mathbf{z})]$, where the true posterior distribution is approximated by a parametric family of posteriors $q_\phi(\mathbf{z}|\mathbf{x})$ with parameters $\phi$. Learning involves joint optimization of the variational parameters $\phi$ and model parameters $\theta$. In this framework, we can think of the generative model as a decoder of the latent variables, and the inference network as an encoder of the observed data into the latent representation. Gradients of $\mathcal{F}$ are estimated using path-wise derivative estimators ('reparameterization trick') [12, 15].

## 2.1 Architectures

We build on recent work on *sequential* generative models [7, 11, 6] by extending them to operate on different 3D representations. This family of models generates the observed data over the course of $T$ computational steps. More precisely, these models operate by sequentially transforming independently generated Gaussian latent variables into refinements of a hidden representation $\mathbf{h}$, which we refer to as the 'canvas'. The final configuration of the canvas, $\mathbf{h}_T$, is then transformed into the target data $\mathbf{x}$ (e.g. an image) through a final smooth transformation. In our framework, we refer to the hidden representation $\mathbf{h}_T$ as the '3D representation' since it will have a special form that is amenable to 3D transformations. This generative process is described by the following equations:

$$\text{Latents } \mathbf{z}_t \sim \mathcal{N}(\cdot|\mathbf{0},\mathbf{1}) \qquad (1) \qquad \text{3D representation } \mathbf{h}_t = f_{\text{write}}(\mathbf{s}_t, \mathbf{h}_{t-1}; \theta_w) \quad (4)$$

$$\text{Encoding } \mathbf{e}_t = f_{\text{read}}(\mathbf{c}, \mathbf{s}_{t-1}; \theta_r) \qquad (2) \qquad \text{2D projection } \hat{\mathbf{x}} = \mathrm{Proj}(\mathbf{h}_T, \mathbf{s}_T; \theta_p) \qquad (5)$$

$$\text{Hidden state } \mathbf{s}_t = f_{\text{state}}(\mathbf{s}_{t-1}, \mathbf{z}_t, \mathbf{e}_t; \theta_s) \quad (3) \qquad \text{Observation } \mathbf{x} \sim p(\mathbf{x}|\hat{\mathbf{x}}). \qquad (6)$$

Each step generates an independent set of $K$-dimensional variables $\mathbf{z}_t$ (equation 1). We use a fully connected long short-term memory network (LSTM, [8]) as the transition function $f_{\text{state}}(\mathbf{s}_{t-1}, \mathbf{z}_t, \mathbf{c}; \theta_s)$. The context encoder $f_{\text{read}}(\mathbf{c}, \mathbf{s}_{t-1}; \theta_r)$ is task dependent; we provide further details in section 3.

When using a volumetric latent 3D representation, the representation update function $f_{\text{write}}(\mathbf{s}_t, \mathbf{h}_{t-1}; \theta_w)$ in equation 4 is parameterized by a volumetric spatial transformer (VST, [9]). More precisely, we set $f_{\text{write}}(\mathbf{s}_t, \mathbf{h}_{t-1}; \theta_w) = \mathrm{VST}(g_1(\mathbf{s}_t), g_2(\mathbf{s}_t))$ where $g_1$ and $g_2$ are MLPs that take the state $\mathbf{s}_t$ and map it to appropriate sizes. More details about the VST are provided in the appendix A.3. When using a mesh 3D representation $f_{\text{write}}$ is a fully-connected MLP.

The function $\mathrm{Proj}(\mathbf{h}_T, \mathbf{s}_T)$ is a projection operator from the model's latent 3D representation $\mathbf{h}_T$ to the training data's domain (which in our experiments is either a volume or an image) and plays the role of a 'renderer'. The conditional density $p(\mathbf{x}|\hat{\mathbf{x}})$ is either a diagonal Gaussian (for real-valued data) or a product of Bernoulli distributions (for binary data). We denote the set of all parameters of this generative model as $\theta = \{\theta_r, \theta_w, \theta_s, \theta_p\}$. Details of the inference model and the variational bound is provided in the appendix A.2.

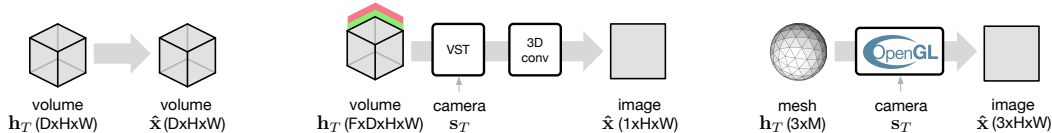

volume
$\mathbf{h}_T$ (DxHxW)

volume
$\hat{\mathbf{x}}$ (DxHxW)

volume
$\mathbf{h}_T$ (FxDxHxW)

camera
$\mathbf{s}_T$

image
$\hat{\mathbf{x}}$ (1xHxW)

mesh
$\mathbf{h}_T$ (3xM)

camera
$\mathbf{s}_T$

image
$\hat{\mathbf{x}}$ (3xHxW)

Figure 3: **Projection operators:** These drop-in modules relate a latent 3D representation with the training data. The choice of representation and the type of available training data determine which operator should be used. *Left:* Volume-to-volume projection (no parameters). *Middle:* Volume-to-image neural projection (learnable parameters). *Right:* Mesh-to-image OpenGL projection (no learnable parameters).

Here we discuss the projection operators in detail. These drop-in modules relate a latent 3D representation with the training data. The choice of representation (volume or mesh) and the type of available training data (3D or 2D) determine which operator is used.

**3D → 3D projection (identity):** In cases where training data is already in the form of volumes (e.g., in medical imagery, volumetrically rendered objects, or videos), we can directly define the likelihood density $p(\mathbf{x}|\hat{\mathbf{x}})$, and the projection operator is simply the identity $\hat{\mathbf{x}} = \mathbf{h}_T$ function (see figure 3 left).

**3D → 2D neural projection (learned):** In most practical applications we only have access to images captured by a camera. Moreover, the camera pose may be unknown or partially known. For these cases, we construct and learn a map from an $F$-dimensional volume $\mathbf{h}_T$ to the observed 2D images by combining the VST with 3D and 2D convolutions. When multiple views from different positions are simultaneously observed, the projection operator is simply cloned as many times as there are target views. The parameters of the projection operator are trained jointly with the rest of the model. This operator is depicted in figure 3 (middle). For details see appendix A.4.

**3D → 2D OpenGL projection (fixed):** When working with a mesh representation, the projection operator in equation 4 is a complex map from the mesh description $\mathbf{h}$ provided by the generative model to the rendered images $\hat{\mathbf{x}}$. In our experiments we use an off-the-shelf OpenGL renderer and treat it as a black-box with no parameters. This operator is depicted in figure 3 (right).

A challenge in working with black-box renderers is that of back-propagating errors from the image to the mesh. This requires either a differentiable renderer [19], or resort to gradient estimation techniques such as finite-differences [5] or Monte Carlo estimators [21, 1]. We opt for a scheme based on REINFORCE [26], details of which are provided in appendix A.5.

## 3 Experiments

We demonstrate the ability of our model to learn and exploit 3D scene representations in five challenging tasks. These tasks establish it as a powerful, robust and scalable model that is able to provide high quality generations of 3D scenes, can robustly be used as a tool for 3D scene completion, can be adapted to provide class-specific or view-specific generations that allow variations in scenes to be explored, can synthesize multiple 2D scenes to form a coherent understanding of a scene, and can operate with complex visual systems such as graphics renderers. We explore four data sets:

**Necker cubes** The Necker cube is a classical psychological test of the human ability for 3D and spatial reasoning. This is the simplest dataset we use and consists of $40 \times 40 \times 40$ volumes with a $10 \times 10 \times 10$ wire-frame cube drawn at a random orientation at the center of the volume [25].

**Primitives** The volumetric primitives are of size $30 \times 30 \times 30$. Each volume contains a simple solid geometric primitive (e.g., cube, sphere, pyramid, cylinder, capsule or ellipsoid) that undergoes random translations ($[0, 20]$ pixels) and rotations ($[-\pi, \pi]$ radians).

**MNIST3D** We extended the MNIST dataset [18] to create a $30 \times 30 \times 30$ volumetric dataset by extruding the MNIST images. The resulting dataset has the same number of images as MNIST. The data is then augmented with random translations ($[0, 20]$ pixels) and rotations ($[-\pi, \pi]$ radians) that are procedurally applied during training.

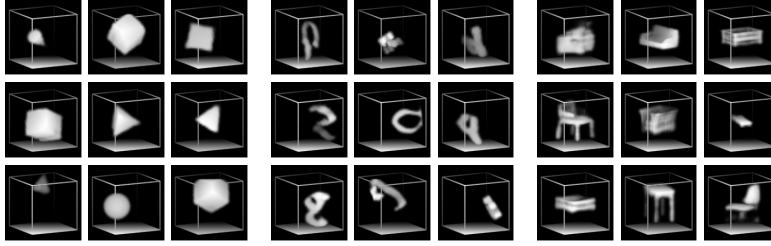

Figure 4: **A generative model of volumes:** For each dataset we display 9 samples from the model. The samples are sharp and capture the multi-modality of the data. *Left:* Primitives (trained with translations and rotations). *Middle:* MNIST3D (translations and rotations). *Right:* ShapeNet (trained with rotations only). Videos of these samples can be seen at `https://goo.gl/9hCkxs`.

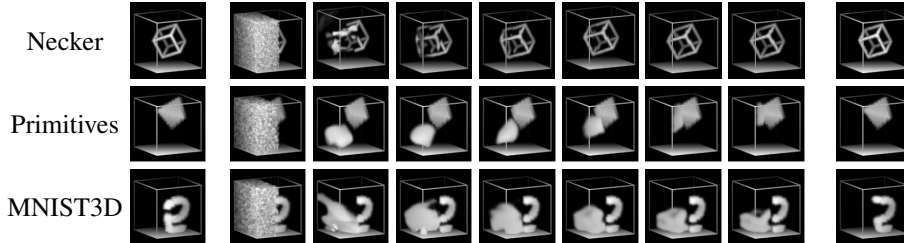

Figure 5: **Probabilistic volume completion (Necker Cube, Primitives, MNIST3D):** *Left:* Full ground-truth volume. *Middle:* First few steps of the MCMC chain completing the missing left half of the data volume. *Right:* 100th iteration of the MCMC chain. Best viewed on a screen. Videos of these samples can be seen at `https://goo.gl/9hCkxs`.

**ShapeNet** The ShapeNet dataset [2] is a large dataset of 3D meshes of objects. We experiment with a 40-class subset of the dataset, commonly referred to as ShapeNet40. We render each mesh as a binary $30 \times 30 \times 30$ volume.

For all experiments we used LSTMs with 300 hidden neurons and 10 latent variables per generation step. The context encoder $f_c(\mathbf{c}, \mathbf{s}_{t-1})$ was varied for each task. For image inputs we used convolutions and standard spatial transformers, and for volumes we used volumetric convolutions and VSTs. For the class-conditional experiments, the context $\mathbf{c}$ is a one-hot encoding of the class. As meshes are much lower-dimensional than volumes, we set the number of steps to be $T = 1$ when working with this representation. We used the Adam optimizer [14] for all experiments.

### 3.1 Generating volumes

When ground-truth volumes are available we can directly train the model using the identity projection operator (see section 2.1). We explore the performance of our model by training on several datasets. We show in figure 4 that it can capture rich statistics of shapes, translations and rotations across the datasets. For simpler datasets such as Primitives and MNIST3D (figure 4 left, middle), the model learns to produce very sharp samples. Even for the more complex ShapeNet dataset (figure 4 right) its samples show a large diversity of shapes whilst maintaining fine details.

### 3.2 Probabilistic volume completion and denoising

We test the ability of the model to impute missing data in 3D volumes. This is a capability that is often needed to remedy sensor defects that result in missing or corrupt regions, (see for instance [29, 4]). For volume completion, we use an unconditional volumetric model and alternate between inference and generation, feeding the result of one into the other. This procedure simulates a Markov chain and samples from the correct distribution, as we show in appendix A.10. We test the model by occluding half of a volume and completing the missing half. Figure 5 demonstrates that our model successfully completes large missing regions with high precision. More examples are shown in the appendix A.7.

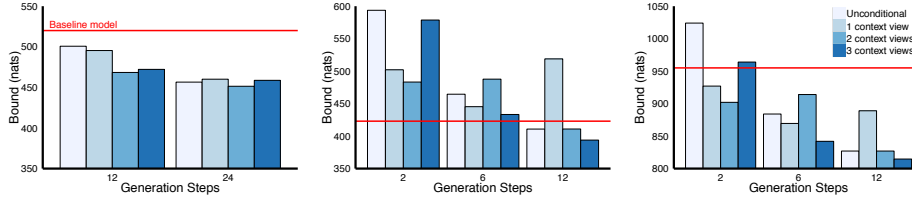

Figure 6: **Quantitative results:** Increasing the number of steps or the number of contextual views both lead to improved log-likelihoods. *Left:* Primitives. *Middle:* MNIST3D. *Right:* ShapeNet.

### 3.3 Conditional volume generation

The models can also be trained with context representing the class of the object, allowing for class conditional generation. We train a class-conditional model on ShapeNet and show multiple samples for 10 of the 40 classes in figure 7. The model produces high-quality samples of all classes. We note their sharpness, and that they accurately capture object rotations, and also provide a variety of plausible generations. Samples for all 40 ShapeNet classes are shown in appendix A.8.

We also form conditional models using a single view of 2D contexts. Our results, shown in figure 8 indicate that the model generates plausible shapes that match the constraints provided by the context and captures the multi-modality of the posterior. For instance, consider figure 8 (right). The model is conditioned on a single view of an object that has a triangular shape. The model's three shown samples have greatly varying shape (e.g., one is a cone and the other a pyramid), whilst maintaining the same triangular projection. More examples of these inferences are shown in the appendix A.9.

### 3.4 Performance benchmarking

We quantify the performance of the model by computing likelihood scores, varying the number of conditioning views and the number of inference steps in the model. Figure 6 indicates that the number of generation steps is a very important factor for performance (note that increasing the number of steps does not affect the total number of parameters in the model). Additional context views generally improves the model's performance but the effect is relatively small. With these experiments we establish the first benchmark of likelihood-bounds on Primitives (unconditional: 500 nats; 3-views: 472 nats), MNIST3D (unconditional: 410 nats; 3-views: 393 nats) and ShapeNet (unconditional: 827 nats; 3-views: 814 nats). As a strong baseline, we have also trained a deterministic 6-layer volumetric convolutional network with Bernoulli likelihoods to generate volumes conditioned on 3 views. The performance of this model is indicated by the red line in figure 6. Our generative model substantially outperforms the baseline for all 3 datasets, even when conditioned on a *single view*.

### 3.5 Multi-view training

In most practical applications, ground-truth volumes are not available for training. Instead, data is captured as a collection of images (e.g., from a multi-camera rig or a moving robot). To accommodate this fact, we extend the generative model with a projection operator that maps the internal volumetric representation $\mathbf{h}_T$ to a 2D image $\hat{\mathbf{x}}$. This map imitates a 'camera' in that it first applies an affine transformation to the volumetric representation, and then flattens the result using a convolutional network. The parameters of this projection operator are trained jointly with the rest of the model. Further details are explained in the appendix A.4.

In this experiment we train the model to learn to reproduce an image of the object given one or more views of it from fixed camera locations. It is the model's responsibility to infer the volumetric representation as well as the camera's position relative to the volume. It is clear to see how the model can 'cheat' by generating volumes that lead to good reconstructions but do not capture the underlying 3D structure. We overcome this by reconstructing *multiple* views from the *same* volumetric representation and using the context information to fix a reference frame for the internal volume. This enforces a consistent hidden representation that generalises to new views.

We train a model that conditions on 3 fixed context views to reproduce 10 simultaneous random views of an object. After training, we can sample a 3D representation given the context, and render it from arbitrary camera angles. We show the model's ability to perform this kind of inference in figure

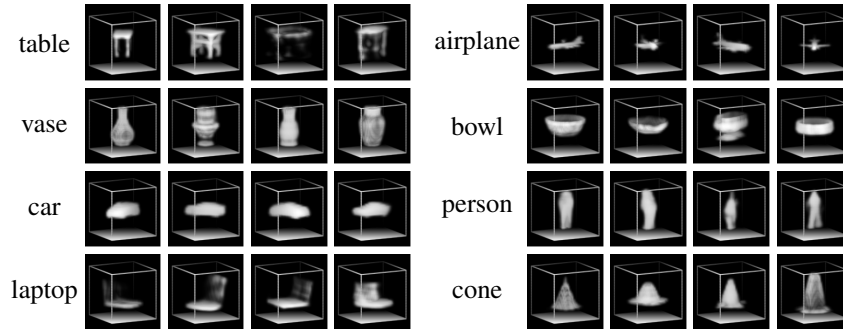

Figure 7: **Class-conditional samples**: Given a one-hot encoding of class as context, the model produces high-quality samples. Notice, for instance, sharpness and variability of generations for 'chair', accurate capture of rotations for 'car', and even identifiable legs for the 'person' class. Videos of these samples can be seen at `https://goo.gl/9hCkxs`.

9. The resulting network is capable of producing an abstract 3D representation from 2D observations that is amenable to, for instance, arbitrary camera rotations.

### 3.6 Single-view training

Finally, we consider a mesh-based 3D representation and demonstrate the feasibility of training our models with a fully-fledged, black-box renderer in the loop. Such renderers (e.g. OpenGL) accurately capture the relationship between a 3D representation and its 2D rendering out of the box. This image is a complex function of the objects' colors, materials and textures, positions of lights, and that of other objects. By building this knowledge into the model we give hints for learning and constrain its hidden representation.

We consider again the Primitives dataset, however now we only have access to 2D images of the objects at training time. The primitives are textured with a color on each side (which increases the complexity of the data, but also makes it easier to detect the object's orientation relative to the camera), and are rendered under three lights. We train an unconditional model that given a 2D image, infers the parameters of a 3D mesh and its orientation relative to the camera, such that when textured and rendered reconstructs the image accurately. The inferred mesh is formed by a collection of 162 vertices that can move on fixed lines that spread from the object's center, and is parameterized by the vertices' positions on these lines.

The results of these experiments are shown in figure 10. We observe that in addition to reconstructing the images accurately (which implies correct inference of mesh and camera), the model correctly infers the extents of the object not in view, as demonstrated by views of the inferred mesh from unobserved camera angles.

## 4 Discussion

In this paper we introduced a powerful family of 3D generative models inspired by recent advances in image modeling. When trained on ground-truth volumes, they can produce high-quality samples that capture the multi-modality of the data. We further showed how common inference tasks, such as that of inferring a posterior over 3D structures given a 2D image, can be performed efficiently via conditional training. We also demonstrated end-to-end training of such models directly from 2D images through the use of differentiable renderers. This demonstrates for the first time the feasibility of learning to infer 3D representations in a purely unsupervised manner.

We experimented with two kinds of 3D representations: volumes and meshes. Volumes are flexible and can capture a diverse range of structures, however they introduce modeling and computational challenges due to their high dimensionality. Conversely, meshes can be much lower dimensional and therefore easier to work with, and they are the data-type of choice for common rendering engines, however standard paramaterizations can be restrictive in the range of shapes they can capture.

It will be of interest to consider other representation types, such as NURBS, or training with a volume-to-mesh conversion algorithm (e.g., marching cubes) in the loop.

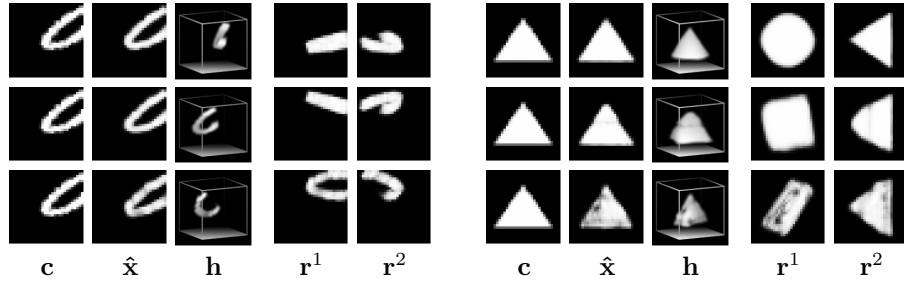

$$\mathbf{c} \qquad \hat{\mathbf{x}} \qquad \mathbf{h} \qquad \mathbf{r}^1 \qquad \mathbf{r}^2 \qquad\qquad \mathbf{c} \qquad \hat{\mathbf{x}} \qquad \mathbf{h} \qquad \mathbf{r}^1 \qquad \mathbf{r}^2$$

Figure 8: **Recovering 3D structure from 2D images:** The model is trained on volumes, conditioned on $\mathbf{c}$ as context. Each row corresponds to an independent sample $\mathbf{h}$ from the model given $\mathbf{c}$. We display $\hat{\mathbf{x}}$, which is $\mathbf{h}$ viewed from the same angle as $\mathbf{c}$. Columns $\mathbf{r}^1$ and $\mathbf{r}^2$ display the inferred 3D representation $\mathbf{h}$ from different viewpoints. The model generates plausible, but varying, interpretations, capturing the inherent ambiguity of the problem. *Left:* MNIST3D. *Right:* ShapeNet. Videos of these samples can be seen at `https://goo.gl/9hCkxs`.

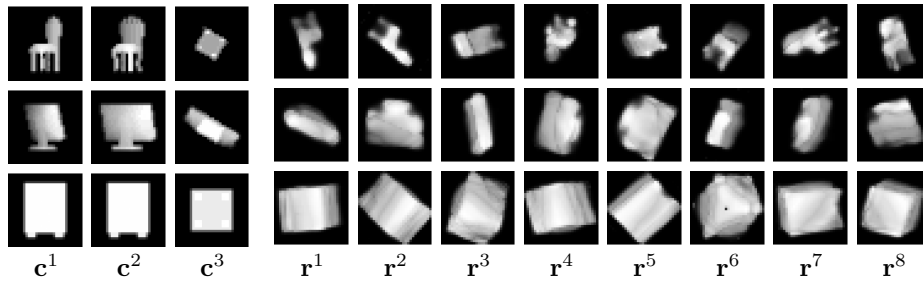

$$\mathbf{c}^1 \quad \mathbf{c}^2 \quad \mathbf{c}^3 \quad \mathbf{r}^1 \quad \mathbf{r}^2 \quad \mathbf{r}^3 \quad \mathbf{r}^4 \quad \mathbf{r}^5 \quad \mathbf{r}^6 \quad \mathbf{r}^7 \quad \mathbf{r}^8$$

Figure 9: **3D structure from multiple 2D images:** Conditioned on 3 depth images of an object, the model is trained to generate depth images of that object from 10 different views. *Left:* Context views. *Right:* Columns $\mathbf{r}^1$ through $\mathbf{r}^8$ display the inferred abstract 3D representation $\mathbf{h}$ rendered from different viewpoints by the learned projection operator. Videos of these samples can be seen at `https://goo.gl/9hCkxs`.

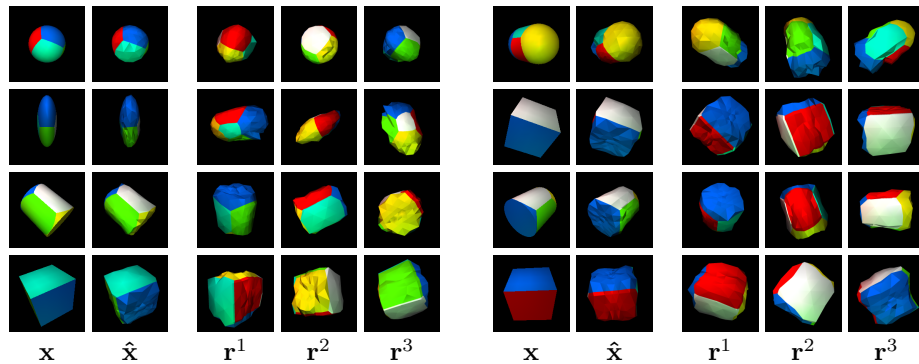

$$\mathbf{x} \qquad \hat{\mathbf{x}} \qquad \mathbf{r}^1 \qquad \mathbf{r}^2 \qquad \mathbf{r}^3 \qquad\qquad \mathbf{x} \qquad \hat{\mathbf{x}} \qquad \mathbf{r}^1 \qquad \mathbf{r}^2 \qquad \mathbf{r}^3$$

Figure 10: **Unsupervised learning of 3D structure:** The model observes $\mathbf{x}$ and is trained to reconstruct it using a mesh representation and an OpenGL renderer, resulting in $\hat{\mathbf{x}}$. We rotate the camera around the inferred mesh to visualize the model's understanding of 3D shape. We observe that in addition to reconstructing accurately, the model correctly infers the extents of the object not in view, demonstrating true 3D understanding of the scene. Videos of these reconstructions have been included in the supplementary material. Best viewed in color. Videos of these samples can be seen at `https://goo.gl/9hCkxs`.

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
