[Supplementary Material]

# Supplementary Material for: Unsupervised Learning of 3D Structure from Images

**Danilo Jimenez Rezende\***
danilor@google.com

**S. M. Ali Eslami\***
aeslami@google.com

**Shakir Mohamed\***
shakir@google.com

**Peter Battaglia\***
peterbattaglia@google.com

**Max Jaderberg\***
jaderberg@google.com

**Nicolas Heess\***
heess@google.com
* Google DeepMind

## A   Appendix

### A.1   Supplementary related work

Volumetric representations have been explored extensively for the tasks of object classification [1, 2, 3, 4], object reconstruction from images [5], volumetric denoising [4, 5] and density estimation [4]. The model we present in this paper extends ideas from the current state-of-the art in deep generative modelling of images [6, 7, 8] to volumetric data. Since these models operate on smooth internal representations, they can be combined with continuous projection operators more easily than prior work.

On the other hand, mesh representations allow for a more compact, yet still rich, representation space. When combined with OpenGL, we can exploit these representations to more accurately capture the physics of the rendering process. Related work include deformable-parts models [9, 10, 11] and approaches from inverse graphics [12, 13, 14, 15].

### A.2   Inference model

We use a structured posterior approximation that has an auto-regressive form, i.e. $q(\mathbf{z}_t|\mathbf{z}_{<t}, \mathbf{x}, \mathbf{c})$. This distribution is parameterized by a deep network:

$$\text{Read Operation} \quad \mathbf{r}_t \quad = f_r(\mathbf{x}, \mathbf{s}_{t-1}; \phi_r) \tag{1}$$

$$\text{Sample} \quad \mathbf{z}_t \quad \sim \mathcal{N}(\mathbf{z}_t|\boldsymbol{\mu}(\mathbf{r}_t, \mathbf{s}_{t-1}, \mathbf{c}; \phi_\mu), \sigma(\mathbf{r}_t, \mathbf{s}_{t-1}, \mathbf{c}; \phi_\sigma)) \tag{2}$$

The 'read' function $f_r$ is parametrized in the same way as $f_w(\mathbf{s}_t, \mathbf{h}_{t-1}; \theta_h)$. During inference, the states $s_t$ are computed using the same state transition function as in the generative model. We denote the parameters of the inference model by $\phi = \{\phi_r, \phi_\mu, \phi_\sigma\}$.

The variational loss function associated with this model is given by:

$$\mathcal{F} = -\mathbb{E}_{q(\mathbf{z}_{1,\ldots,T}|\mathbf{x},\mathbf{c})}[\log p_\theta(\mathbf{x}|\mathbf{z}_{1,\ldots,T}, \mathbf{c})] + \sum_{t=1}^{T} \text{KL}[q_\phi(\mathbf{z}_t|\mathbf{z}_{<t}\mathbf{x})\|p(\mathbf{z}_t)], \tag{3}$$

where $\mathbf{z}_{<t}$ indicates the collection of all latent variables from iteration 1 to $t-1$. We can now optimize this objective function for the variational parameters $\phi$ and the model parameters $\theta$ by stochastic gradient descent.

### A.3 Volumetric Spatial Transformers

Spatial transformers [16] provide a flexible mechanism for smooth attention and can be easily applied to both 2 and 3 dimensional data. Spatial Transformers process an input image $\mathbf{x}$, using parameters $\mathbf{h}$, and generate an output $\mathrm{ST}(\mathbf{x}, \mathbf{h})$:

$$\mathrm{ST}(\mathbf{x}, \mathbf{h}) = [\kappa_h(\mathbf{h}) \otimes \kappa_w(\mathbf{h})] * \mathbf{x},$$

where $\kappa_h$ and $\kappa_w$ are 1-dimensional kernels, $\otimes$ indicates the tensor outer-product of the three kernels and $*$ indicates a convolution. Similarly, Volumetric Spatial Transformers (VST) process an input data volume $\mathbf{x}$, using parameters $\mathbf{h}$, and generate an output $\mathrm{VST}(\mathbf{x}, \mathbf{h})$:

$$\mathrm{VST}(\mathbf{x}, \mathbf{h}) = [\kappa_d(\mathbf{h}) \otimes \kappa_h(\mathbf{h}) \otimes \kappa_w(\mathbf{h})] * \mathbf{x},$$

where $\kappa_d$, $\kappa_h$ and $\kappa_w$ are 1-dimensional kernels, $\otimes$ indicates the tensor outer-product of the three kernels and $*$ indicates a convolution. The kernels $\kappa_d$, $\kappa_h$ and $\kappa_w$ used in this paper correspond to a simple affine transformation of a 3-dimensional grid of points that uniformly covers the input image.

## A.4 Learnable $3D \rightarrow 2D$ projection operators

These projection operators or 'learnable cameras' are built by first applying a affine transformation to the volumetric canvas $\mathbf{c}_T$ using the Spatial Transformer followed a combination on 3D and 2D convolutions as depicted in figure A.4.

Figure 1: **Learnable projection operators:** Multiple instances of the projection operator (with shared parameters).

## A.5 Stochastic Gradient Estimators for Expectations of Black-Box Functions

We employ a multi-sample extension of REINFORCE, inspired by [17, 18]. For each image we sample $K$ realizations of the inferred mesh for a fixed set of latent variables using a small Gaussian noise and compute its corresponding render. The variance of the learning signal for each sample $k$ is reduced by computing a 'baseline' using the $K-1$ remaining samples. See [17] for further details. The estimator is easy to implement and we found this approach to work well in practice even for relatively high-dimensional meshes.

## A.6  Unconditional generation

In figures 2 and 3 we show further examples of our model's capabilities at unconditional volume generation.

Figure 2: **A strong generative model of volumes (Primitives):** *Left:* Examples of training data. *Right:* Samples from the model.

Figure 3: **A strong generative model of volumes (MNIST3D):** *Left:* Examples of training data. *Right:* Samples from the model.

## A.7 Volume completion

In figures 4, 5 and 6 we show the model's capabilities at volume completion.

Figure 4: **Completion using a model of 3D trained on volumes (Necker cube):** *Left:* Full target volume. *Middle:* First 8 steps of the MCMC chain completing the missing left half of the data volume. *Right:* 100th iteration of the MCMC chain.

Figure 5: **Completion using a model of 3D trained on volumes (Primitives):** *Left:* Full target volume. *Middle:* First 8 steps of the MCMC chain completing the missing left half of the data volume. *Right:* 100th iteration of the MCMC chain.

Figure 6: **Completion using a model of 3D trained on volumes (MNIST3D):** *Left:* Full target volume. *Middle:* First 8 steps of the MCMC chain completing the missing left half of the data volume. *Right:* 100th iteration of the MCMC chain.

## A.8 Class-conditional volume generation

In figure 7 we show samples from a class-conditional volumetric generative model for all 40 ShapeNet classes.

Figure 7: **Class-Conditional Volumetric Generation (ShapeNet):** All 40 classes.

## A.9 View-conditional volume generation

In figures 8, 9 and 10 we show samples from a view-conditional volumetric generative model for Primitives, MNIST3D and ShapeNet respectively.

Figure 8: **Recovering 3D structure from 2D images (Primitives):** The model is trained on volumes, conditioned on $\mathbf{c}$ as context. Each row corresponds to an independent sample $\mathbf{h}$ from the model given $\mathbf{c}$. We display $\hat{\mathbf{x}}$, which is $\mathbf{h}$ viewed from the same angle as $\mathbf{c}$. Columns $\mathbf{r}^1$ and $\mathbf{r}^2$ display the inferred 3D representation $\mathbf{h}$ from different viewpoints. The model generates plausible, but varying, interpretations, capturing the inherent ambiguity of the problem.

Figure 9: **Recovering 3D structure from 2D images (MNIST3D):** The model is trained on volumes, conditioned on $\mathbf{c}$ as context. Each row corresponds to an independent sample $\mathbf{h}$ from the model given $\mathbf{c}$. We display $\hat{\mathbf{x}}$, which is $\mathbf{h}$ viewed from the same angle as $\mathbf{c}$. Columns $\mathbf{r}^1$ and $\mathbf{r}^2$ display the inferred 3D representation $\mathbf{h}$ from different viewpoints. The model generates plausible, but varying, interpretations, capturing the inherent ambiguity of the problem.

Figure 10: **Recovering 3D structure from 2D images (ShapeNet):** The model is trained on volumes, conditioned on $\mathbf{c}$ as context. Each row corresponds to an independent sample $\mathbf{h}$ from the model given $\mathbf{c}$. We display $\hat{\mathbf{x}}$, which is $\mathbf{h}$ viewed from the same angle as $\mathbf{c}$. Columns $\mathbf{r}^1$ and $\mathbf{r}^2$ display the inferred 3D representation $\mathbf{h}$ from different viewpoints. The model generates plausible, but varying, interpretations, capturing the inherent ambiguity of the problem.

## A.10 Volume completion with MCMC

When only part of the data-vector $\mathbf{x}$ is observed, we can approximately sample the missing part of the volume conditioned on the observed part by building a Markov Chain. We review below the derivations from [19] for completeness. Let $\mathbf{x}_o$ and $\mathbf{x}_u$ be the observed and unobserved parts of $\mathbf{x}$ respectively. The observed $\mathbf{x}_o$ is fixed throughout, therefore all the computations in this section will be conditioned on $\mathbf{x}_o$. The imputation procedure can be written formally as a Markov chain on the space of missing entries $\mathbf{x}_u$ with transition kernel $\mathcal{K}^q(\mathbf{x}'_u|\mathbf{x}_u,\mathbf{x}_o)$ given by

$$\mathcal{K}^q(\mathbf{x}'_u|\mathbf{x}_u,\mathbf{x}_o) = \iint p(\mathbf{x}'_u,\mathbf{x}'_o|\mathbf{z})q(\mathbf{z}|\mathbf{x})d\mathbf{x}'_o d\mathbf{z}, \tag{4}$$

where $\mathbf{x} = (\mathbf{x}_u,\mathbf{x}_o)$.

Provided that the recognition model $q(\mathbf{z}|\mathbf{x})$ constitutes a good approximation of the true posterior $p(\mathbf{z}|\mathbf{x})$, (4) can be seen as an approximation of the kernel

$$\mathcal{K}(\mathbf{x}'_u|\mathbf{x}_u,\mathbf{x}_o) = \iint p(\mathbf{x}'_u,\mathbf{x}'_o|\mathbf{z})p(\mathbf{z}|\mathbf{x})d\mathbf{x}'_o d\mathbf{z}. \tag{5}$$

The kernel (5) has two important properties: (i) it has as its eigen-distribution the marginal $p(\mathbf{x}_u|\mathbf{x}_o)$; (ii) $\mathcal{K}(\mathbf{x}'_u|\mathbf{x}_u,\mathbf{x}_o) > 0\ \forall \mathbf{x}_o,\mathbf{x}_u,\mathbf{x}'_u$. The property (i) can be derived by applying the kernel (5) to the marginal $p(\mathbf{x}_u|\mathbf{x}_o)$ and noting that it is a fixed point. Property (ii) is an immediate consequence of the smoothness of the model.

We apply the fundamental theorem for Markov chains and conclude that given the above properties, a Markov chain generated by (5) is guaranteed to generate samples from the correct marginal $p(\mathbf{x}_u|\mathbf{x}_o)$.

In practice, the stationary distribution of the completed data will not be exactly the marginal $p(\mathbf{x}_u|\mathbf{x}_o)$, since we use the approximated kernel (4). Even in this setting we can provide a bound on the $L_1$ norm of the difference between the resulting stationary marginal and the target marginal $p(\mathbf{x}_u|\mathbf{x}_o)$

**Proposition A.1** ($L_1$ bound on marginal error ). *If the recognition model $q(\mathbf{z}|\mathbf{x})$ is such that for all $\mathbf{z}$*

$$\exists \varepsilon > 0 \ s.t. \ \int \left| \frac{q(\mathbf{z}|\mathbf{x})p(\mathbf{x})}{p(\mathbf{z})} - p(\mathbf{x}|\mathbf{z}) \right| d\mathbf{x} \le \varepsilon \tag{6}$$

*then the marginal $p(\mathbf{x}_u|\mathbf{x}_o)$ is a weak fixed point of the kernel (4) in the following sense:*

$$\int \left| \int \left( \mathcal{K}^q(\mathbf{x}'_u|\mathbf{x}_u,\mathbf{x}_o) - \mathcal{K}(\mathbf{x}'_u|\mathbf{x}_u,\mathbf{x}_o) \right) p(\mathbf{x}_u|\mathbf{x}_o)d\mathbf{x}_u \right| d\mathbf{x}'_u < \varepsilon. \tag{7}$$

*Proof.*

$$\int \left| \int [\mathcal{K}^q(\mathbf{x}'_u|\mathbf{x}_u,\mathbf{x}_o) - \mathcal{K}(\mathbf{x}'_u|\mathbf{x}_u,\mathbf{x}_o)] p(\mathbf{x}_u|\mathbf{x}_o)d\mathbf{x}_u \right| d\mathbf{x}'_u$$

$$= \int | \iint p(\mathbf{x}'_u,\mathbf{x}'_o|\mathbf{z})p(\mathbf{x}_u,\mathbf{x}_o)[q(\mathbf{z}|\mathbf{x}_u,\mathbf{x}_o)$$

$$- p(\mathbf{z}|\mathbf{x}_u,\mathbf{x}_o)]d\mathbf{x}_u d\mathbf{z} | d\mathbf{x}'_u$$

$$= \int \left| \int p(\mathbf{x}'|\mathbf{z})p(\mathbf{x})[q(\mathbf{z}|\mathbf{x}) - p(\mathbf{z}|\mathbf{x})] \frac{p(\mathbf{x})}{p(\mathbf{z})}\frac{p(\mathbf{z})}{p(\mathbf{x})}d\mathbf{x}d\mathbf{z} \right| d\mathbf{x}'$$

$$= \int \left| \int p(\mathbf{x}'|\mathbf{z})p(\mathbf{z})[q(\mathbf{z}|\mathbf{x})\frac{p(\mathbf{x})}{p(\mathbf{z})} - p(\mathbf{x}|\mathbf{z})]d\mathbf{x}d\mathbf{z} \right| d\mathbf{x}'$$

$$\le \int \int p(\mathbf{x}'|\mathbf{z})p(\mathbf{z}) \int \left| q(\mathbf{z}|\mathbf{x})\frac{p(\mathbf{x})}{p(\mathbf{z})} - p(\mathbf{x}|\mathbf{z}) \right| d\mathbf{x}d\mathbf{z}d\mathbf{x}'$$

$$\le \varepsilon,$$

where we apply the condition (6) to obtain the last statement. $\qquad \square$

That is, if the recognition model is sufficiently close to the true posterior to guarantee that (6) holds for some acceptable error $\varepsilon$ than (7) guarantees that the fixed-point of the Markov chain induced by the kernel (4) is no further than $\varepsilon$ from the true marginal with respect to the $L_1$ norm.