[Reviews · NeurIPS 2016]

Reviewer 1

Summary

This is an interesting paper combining autoencoders, 3D representation, and LSTMs for learning neural networks with 3D representation. Learning 3D representation is important, which makes this paper interesting. More experiments into the reason for selecting some of the architectures could be useful.

Qualitative Assessment

This is an interesting paper combining autoencoders, 3D representation, and LSTMs for learning neural networks with 3D representation. Learning 3D representation is important, which makes this paper interesting. some 'tricks' used by this paper is useful and interesting. For example. using a 3D+2D convolutional network instead of a render makes the decoder differentiable. modeling only the distance of 162 vertices from the centroid of the object makes predicting degenerate meshes impossible. The videos are interesting and showcases the capability of 3D representation in a neural network. Some clarifications would be nice: - How important is it to have sequential generative model? What happens if the 3d volume h is generated by a 3D convolutional network instead of a read/write LSTM? - From what distributions do you sample the 2D meshes before using OpenGL to render into an image x? - In the mesh rendering method, did it take many samples per mesh for learning to be stable? - What kind of lighting model used for generating the training data?

Confidence in this Review

2-Confident (read it all; understood it all reasonably well)


Reviewer 2

Summary

This paper demonstrates a model that possesses an understanding of the 3-dimensional world, even when the training data itself consists of 2-dimensional images of 3D objects. The approach is to train the model to learn a latent representation that captures complicated 3D properties, and from which 2D views of the world can be generated that are consistent with the training data. The model itself is a variant of a sequential generative network that is trained via a variational lower bound. The model maintains an internal hidden state that maps onto an internal 3D representation (the canvas) that can be used to generate a 2D image by way of a renderer. There are many ways to represent 3D surfaces and this paper considers a volumetric representation and a mesh representation. The model is demonstrated on numerous applications including generation, conditional generation, imputation and inferring 3D structure from multiple 2D views.

Qualitative Assessment

This paper is primarily about piecing together existing techniques, utilizing the advances in generative modelling in order to properly model 3D structure. Inferring 3D structure from 2D images is a canonical goal of computer vision, but I haven't seen many attempts to solve this in the recent deep learning literature (i.e., in an end-to-end fashion). I think that this is an exciting frontier that will no doubt receive a lot of attention. I think that the main contribution of this paper toward these goals are several interesting datasets and corresponding baselines, which are essential in order to garner mainstream attention. The paper itself is well written, albeit quite dense due to the number of different components that must be included to introduce the problem, model, and applications. This is reflected in the size of the appendix. Even still, I think that the level of technical detail is somewhat light, and a lot of details that would be required for replication are left out. These include specific parameterizations and architectures (e.g., of the VST network or rendering MLP), training parameters, etc. I'm reasonably confident that upon release of these datasets that researchers will be able to meet the baseline numbers, but it would be much easier to reproduce with more specific details. The results themselves are impressive, however I have no baseline for just how difficult the problem is (although I assume it is difficult). That is, there is a lack of a naive baseline that does not involve a sophisticated recurrent generative model. Even a non-recurrent generative model when ground-truth information is available would be interesting. Aside from these minor points I think it is an excellent paper.

Confidence in this Review

2-Confident (read it all; understood it all reasonably well)


Reviewer 3

Summary

The paper proposes a DRAW-like (https://arxiv.org/abs/1502.04623) neural network for 3D input spaces that learns in an unsupervised fashion to generate 3D structures, in some cases purely from learning from 2D views of the data. The propose a new dataset MNIST3D to do such experiments, as well as use the ShapeNet dataset. They show that training class-conditional models on ShapeNet recovers object structures of chairs, persons etc. in a way we expect. They say that the datasets will be available upon publication of the paper.

Qualitative Assessment

Apart from the fact that the paper does not have any baseline models whatsoever, I dont see another flaw in the paper. But the fact that it has no baseline models (simple ones, or ones based on previous literature) is not acceptable. On the section of Potential impact or usefulness, I did not see the original DRAW paper be picked up with excitement by the community since it's publication, though everyone thought it was cool. This paper is DRAW3D, hence, I only gave this a 3 and not a 4 rating.

Confidence in this Review

2-Confident (read it all; understood it all reasonably well)


Reviewer 4

Summary

The paper proposes to learn 3D structure representations in an unsupervised fashion from either 2D images or 3D volumes via sequential generative models. The overall training follows the variational auto-encoder style. From a 3D structure representation learned, two methods are prosed for projecting it onto a 2D image: via 1) a learned network, 2) OpenGL renderer. The models are demonstrated on several datasets: Necker Cube, Primitives, MNIST3D and ShapeNet. The paper shows qualitative results and also quantify their method in log-likelihoods.

Qualitative Assessment

OVERALL: This is a fascinating paper that applies sequential generative models to learn deep representations of 3D structures in an unsupervised manner. I really enjoy reading the paper! It's also a plus that the authors are making the code available for the community. TECHNICAL QUALITY: - The paper proposes a set of well thought out experiments. I particularly like the choice of the datasets (especially the newly created MNIST3D), which are diverse and suitable for the 3D structure generating task. - I wonder if the authors could quantify how much variability (multi-modality) of the data is captured by the model. - For completeness, the paper should also discuss/show the possibility (if any) of overfitting/memorizing the dataset. NOVELTY: The paper develops upon the previous sequential generative models. However, the work is the first to learn to infer 3D representations in an unsupervised way. The proposed learning framework that supports both 3D volumes and 2D images is also novel, afaik. IMPACT: The work will enable unsupervised learning of 3D structures in many different settings, e.g. robots with RGB-D cameras automatically collecting and learning from real-world 3D objects; or learning in 3D simulation, etc. CLARITY & PRESENTATION: - The paper is very well written, and a pleasure to read. I really like Fig. 2. - A minor point: the diversity of generated 3D structures is not easy for one to qualitatively evaluate in black/white figures (e.g. Fig. 7).

Confidence in this Review

2-Confident (read it all; understood it all reasonably well)


Reviewer 5

Summary

This paper provides a generative model of 3D volumes and 2D images using an internal 3D latent representation. The model can be conditioned on a context, which in turn can be either a category or multiple views of the 3D volume. In a very similar way to DRAW, the model transforms a set of latent Gaussian variables into an internal 3D representation, which in turn is converted to a 2D image via a convolutional network or OpenGL renderer. Inference in this generative model is performed using variational inference with a recognition network. Five different tasks are considered.

Qualitative Assessment

The paper is well written, clear and contains appropriate references. Also the number of figures help understand and illustrate the text appropriately. The technical content of the paper seems sound, but its ideas are not particularly novel. The proposed model is an extension of DRAW, which now includes a context (which is observed and used as an additional input to the recognition and generation networks) and generates 3D representations instead of 2D representations. This requires the addition of a projection step to 2D at the end of the generation process when the observations are 2D. Well-known variational inference is applied to the resulting model. Some of the claims in the introduction are misleading. "We show how the aforementioned models and  inference networks can be trained end-to-end directly from 2D images without any use of ground-truth 3D labels.". This make it seem as if 3D latent structure was discovered exclusively from 2D information. This is not the case, and prior knowledge about 3D structure and projection from 3D to 2D is included in every version of the model. Only the last experiment seems not to use 3D input data directly (the multi-view experiment uses depth in the provided context). But in that experiment, an OpenGL renderer is introduced in the loop, providing, if we wish, an infinite amount of 3D ground truth by sampling from its input-output pairs. Additionally, not every 3D shape can be accommodated by the model of the last experiment due to its particular parameterization, which is not pointed out in the paper. I would like more clarity regarding training and test splits. For instance, the objects in Figures 9 and 10 were observed at all during training or not? I would like to distinguish generalization to new perspectives from generalization to entirely new shapes. One thing I found concerning about this paper is the lack of comparison with benchmarks. The experiments comprise 5 different tasks. I would like to see the performance of a reasonable benchmark for each of these tasks, so that it is possible to judge whether the results produced by this model are particularly impressive or not. - For instance, Section 3.5 discusses multiview contexts. The text is not clear about the nature of the views, but the corresponding figure and legend clarify that the input views are 3D. It seems like it should be relatively easy to infer a 3D representation from multiple views with depth information. The obtained results should be compared with those of a different technique for the same task to ascertain whether the model is working particularly well or not. - Similarly Section 3.6 uses a very specific parameterization of the input space that restricts the applicability of the model. My understanding is that the same experiment could not be performed with the objects from ShapeNet, for instance. This should be pointed out. Are the objects we see in Figure 10 used to train the model? If they are as I suspect, couldn't the model be basically memorizing 3D structures for each object and then inferring pose for each test image? (as opposed to being able to get 3D understanding of a new object). Then the problem could be posed as, given a set of training images, identify same-object clusters and find its 3D representation. A baseline for this task should be included so that the quality of the proposed model could be judged. - Section 3.3 and Figure 7: Again the generalization capabilities are unclear. Is the model generating _new_ shapes within the same category or multiple rotations of already-observed shapes? The nearest training neighbor to each generation, computed using a meaningful metric (that matches shape regardless of rotation, translation, and scaling) could be useful here. Minor: Section A.3: "of the three kernels": there are only 2 kernels in the 2D ST. Section 2.1: "fwrite(st,h_t-1;theta_w) = VST(g1(st),g2(st))" This equation doesn't seem correct, isn't h_t-1 used during writing?

Confidence in this Review

3-Expert (read the paper in detail, know the area, quite certain of my opinion)


Reviewer 6

Summary

This paper proposed a generative model of 3D objects. The model is essentially a variational auto-enconder which can be learned directly from 2D images.

Qualitative Assessment

This paper is the first attempt for literally realizing 'vision as inverse graphics'. A highly original and inspiring work. However, there are several issues: 1) Since the title is 'Unsupervised Learning of 3D Structure from Images', I suggest the authors trim or remove the experiments with 3D structure as inputs. Please elaborate more on the learning with only 2D images. For example, what does the model represent when the 2D input image in ambiguous such as the one in Figure 1? 2) In Figure 2 and 3, it's better to change 'OpenGL' to 'renderer'. The proposed method is not restricted to any specific renderer such OpenGL. One could use other renderers such as DirectX. 3) For optimizing a black-box generative model, I refer the authors to Bayesian Optimization for Likelihood-Free Inference of Simulator-Based Statistical Models, by M.U. Gutmann and J. Corander, JMLR 2015.

Confidence in this Review

2-Confident (read it all; understood it all reasonably well)